# Hot Executive Function Assessment Instruments in Preschool Children: A Systematic Review

**DOI:** 10.3390/ijerph19010095

**Published:** 2021-12-23

**Authors:** Vannia Mehsen, Lilian Morag, Sergio Chesta, Kristol Cleaton, Héctor Burgos

**Affiliations:** Escuela de Psicología, Facultad de Ciencias, Universidad Mayor, Temuco 4801043, Chile; v.mehsen@gmail.com (V.M.); lilianmoragm@gmail.com (L.M.); sergio.chesta@umayor.cl (S.C.); kristol.cleaton@mayor.cl (K.C.)

**Keywords:** hot executive functions, assessment instruments, test, preschoolers

## Abstract

The study aimed to systematically analyze the empirical evidence that is available concerning batteries, tests or instruments that assess hot executive functions (EFs) in preschoolers, identifying which are the most used instruments, as well as the most evaluated hot EFs. For the review and selection of articles, the systematic review methodology PRISMA was used. The article search considered the EBSCO, Web of Science (WoS), SciELO and PubMed databases, with the keywords “Hot executive function”, “Assessment”, “test”, “evaluation”, using the Boolean operators AND and OR indistinctly, between 2000 and April 2021. Twenty-four articles were selected and analyzed. The most commonly used instruments to assess hot EFs in preschool children were the Delayed Gratification Task, the Child’s Play Task, and the Delayed Reward Task. Amongst those analyzed, 17 instruments were found to assess hot EFs in preschoolers. The accuracy and conceptual clarity between the assessment of cognitive and emotional components in EFs is still debatable. Nevertheless, the consideration of affective temperature and reward stimulus type, could be an important influence when assessing EFs in this age range. Evidence of the possible involvement of cortical and subcortical structures, as well as the limbic system, in preschool executive functioning assessment has also been incorporated.

## 1. Introduction

This study is framed in the context of neuropsychology, a clinical discipline pertaining to a part of neuroscience that focuses on a neural interpretation of the behavioral and affective cognitive evidence in people [1,2]. Along these lines, the mnesic, attentional, and executive functions (EFs), which is the particular interest of this study, are evaluated according to a person’s response in relation to a possible neurological correlate.

Executive function (EF) is a concept incorporated within the discipline of neuropsychology [3,4]. Alexander Luria associated the prefrontal lobe with the control of superior intellectual activity, whereas the term Executive Functions (EFs) was attributed to Muriel Lezak [5]. EFs are defined as a set of neurocognitive skills that can inhibit, regulate and/or plan behavior, emotional behavior, and complex social functioning, contributing to adaptability, and orienting goal-directed behavior in the individual. Through these processes, proactive, autonomous, and productive activities can be developed [2,5,6,7,8,9].

In recent decades there has been significant interest in the evaluation of EFs during childhood since they contribute to school and social development, and their dysfunction being the cause of some psychopathological and behavioral disorders [2,10,11,12]. Therefore, preschool is considered a period of greater sensitivity in executive development [13], linked to the maturation of cortical brain regions [14,15].

EFs can be classified into cold (only cognitive components) and hot (with socio-emotional components). The former requires logical and critical analysis, involving conscious control of thoughts and actions, such as planning and cognitive flexibility, where preferentially the dorsolateral prefrontal cortex is involved [16]. On the other hand, hot EFs incorporate the socioemotional domain and can be evoked by motivational and emotionally meaningful contexts [2,17,18]. Such domains include emotional regulation, empathy, self-awareness, and spatial adaptation [4], as well as the ability to delay gratification, emotion management, and affective decision making [19]. This implies critical engagement of brain areas such as the anterior prefrontal area, dorsolateral area, cingulate area, and supraorbital area [20,21]. Nevertheless, the boundary between hot and cold EFs is under discussion since both are intimately related to adaptive functions [22].

Regarding theoretical models of EF processing, the Iterative Reprocessing (IR) Model [23] and the Somatic Marker Model [24,25] stand out. The first model posits that EFs modulate attention and control behavior, allowing behavior to be more adaptive, planned, and focused on problem solving [26,27]. It involves the neural processes in cortical structures requiring complex circuits that underlie cognitive processes. These processes are associated with reflection, and lead to the development of specific skills, establishing a continuum between hot and cold EFs [2]. The second neurocognitive model allows framing the impact of emotions on decision making [24,25] through somatic markers. These markers are elaborated in front of important stimuli experienced by the individual and subsequently evoked through similar stimuli. Furthermore, they organize actions by virtue of future outcomes, linked to positive or negative valences that attract attention to relevant stimuli and allow the elaboration of novel action sequences, according to the demand of the context [28]. Thus, this model contributes to the support of cognitive processes, such as advantageous decision making, and allows the person to perform appropriate social behavior [24,25].

In relation to EF assessment, there is a predominance of instruments that examine cognitive skills in general although there are discussions regarding their methodological approach with respect to various theoretical currents [29]. Among the most commonly used instruments to assess EFs are the Wisconsin test and Trail Making Test, and less frequently, the Hayling Test, Iowa Gambling Task (IGT), and Tower of London/Hanoi [30,31]. Fundamentally, these evaluative instruments have been designed for cold EFs, while there is a lack in hot EF evaluations; furthermore, there is less evidence in research related to children and young people, since adult participants seem to be the most studied population in the application of EF instruments [31].

Given that research on hot EFs is scarce, especially when regarding assessment instruments, this review aims to systematically analyze the empirical evidence available pertaining to batteries, tests, or instruments that assess hot EFs in preschool children, as a contribution to the knowledge of the most commonly used instruments when assessing hot executive functions. The intention of this study is to acknowledge key neurocognitive-affective functions in child development as possible educational and socioemotional skill predictors at an early age. This research expects to contribute to a greater interest in promoting these neurocognitive functions in preschool education, with emphasis on their evaluation and stimulation, to develop a better academic, cognitive and socioemotional performance.

## 2. Materials and Methods

The article review was based on international PRISMA statements [32]. The article selection process began with the identification of articles in the Web of Science (WoS), EBSCO, SciELO and PubMed databases, using the keywords in Spanish and English “Función ejecutiva cálida”, “Evaluación”, “Prueba”; “Hot executive function”, “Assessment”, “test”, “evaluation”, with the Boolean operators AND and OR as appropriate, between the periods 2000 to 2020 in April 2021. The review process was confirmed by a second reviewer to ensure the inclusion validity. Differences in opinion were discussed, reaching consensus on the inclusion or exclusion of the study. The methodological quality of the reviewers for each review, as well as the evidence, was assessed using PRISMA.

The review of articles is based on the Preferred Reporting Items for Systematic Reviews and Meta-Analyses (PRISMA) considering selected articles that refer to the effect of interventions on the topic of warm EFs [33,34].

A total of 118 articles were found in WoS, 54 in EBSCO, 16 in SciELO, and 4 in PubMed, for a total of 192 results. The inclusion criteria consisted of (i) strictly empirical research, (ii) a primary focus on Hot EF assessment instruments, (iii) studies done only with a preschool age range, and (iv) publications occurring between 2000 and 2020. The exclusion criteria pertain to (i) systematic reviews or meta-analyses, (ii) articles presenting data only on Cold Executive Functions, and (iii) articles not including a preschool aged sample. 

Of the 192 results, 59 duplicates were eliminated among the four databases, leaving 133 articles. Thereafter, 109 articles were discarded for being systematic reviews (*n* = 13), meta-analyses (*n* = 3), for not including instruments that evaluate hot EFs (*n* = 38) and for not including a preschool aged sample (*n* = 55) (See Figure 1).

## 3. Results

Between the years 2000 and April 2021, there exist 24 articles related to the topic of hot EFs in preschoolers, mainly focused from 2011 to 2020 (*n* = 22; 91.7%), where 87.5% (*n* = 21) evaluate hot and cold EFs. Only three articles exclusively evaluate hot EFs (12.5%). The 24 selected publications use mixed samples and incorporate at least one assessment instrument for this neuropsychological function, most of them coming from WoS (*n* = 20; 83.3%), followed by EBSCO (*n* = 11; 45.8%), PubMed (*n* = 2; 8.3%) and SCIELO (*n* = 1; 4.2%) (See Figure 1).

Research seems to be predominantly done in the USA (*n* = 7; 29.2%), then Canada (*n* = 3; 12.5%), and Italy, the Netherlands, and Poland combined (*n* = 2; 8.3%). Some variables have been associated with altered hot EFs in preschool children, such as preterm birth (*n* = 2; 8.3%), obesity (*n* = 1; 4.2%), attention deficit disorder (*n* = 1; 4.2%), and developmental coordination disorder (*n* = 1; 4.2%). Among the most used instruments with hot EFs in preschool children are the Delay of Gratification Task (*n* = 9; 37.8%), followed by Children’s Gambling Task (*n* = 6; 25%), and Gift Delay Task (*n* = 6; 25%), totaling 17 instruments. The most studied hot EFs were Gratification Delay (*n* = 16; 66.7%), Decision Making (*n* = 7; 29.2%) and Emotional Self-Regulation (*n* = 4; 16.7). The list of instruments that were considered, together with the function evaluated and psychometric aspects that account for reliability and validity aspects, are shown in Table 1. It can be seen that some studies refer exclusively to hot EF aspects and others to the combination of cold and hot EFs.

The hot EF assessment instruments reviewed, for the most part, have an individual format. Two relevant classifications were identified: the first being related to decision making by means of a card game (IGT and ChGT), and the second being related to a gratification delay while waiting to be rewarded (LMT, DGT, GDT, SDT, PSRA, MIDA, CDT, and SwT). Assessment instruments typically use rewards in their application, among them, those which incorporate food (LMT, DGT, DCCS, SDT, PSRA, and CDT), stickers (SS, DGT), coins (DGT), gifts (GDT), and toys (PSRA), were distinguished. The aim of this article is to systematically analyze the empirical evidence available on the types of batteries, tests, or instruments that assess hot EFs in preschool children. In summary, 17 evaluative tools published in 24 articles were reported (See Table 2).

As can be observed in Table 2, the DGT test and PSRA battery stand out as being the most frequently used instruments with measuring gratification delay since they involve considering or waiting for a greater reward. Others, such as CDT and SDT, wait for the sound of a bell to receive a reward. Whereas, with computational applications, games that include prizes, versus waiting for gratification, have also been used. An example of such being MIDA which is more commonly used in children over 5 years of age. Gestures made when approaching a more entertaining toy are also measured using tests such as SwT. Questionnaires that are completed by external observers, such as teachers or parents, where they check the access gratification control to EFIn, are also used. Others deal with attentional inhibition to a desirable object while performing tasks, which can be seen in tests such as DCCS and GDT. ERC and the EM-FIST Tests evaluate affective flexibility in the performance of a task with the influence of a significant character. There are also those in which rewards are given when the least advantageous one is chosen, such as the LMT and the SS, a situation that is referred to as inverse reward in the case of the former. The IGT and ChGT use cards where people evaluate those that have something advantageous such as smiley faces.

## 4. Discussion

The revised articles evaluate hot and cold EFs, covering emotional and cognitive aspects respectively, noting distinct definitions concerning hot and cold EFs with a slight difference between them [82]. For some authors, EFs are a set of neurocognitive abilities that allow adaptability and generate goal-directed behaviors, including emotional and cognitive aspects [5,6,7,8,9]. Yet, according to other authors, this concept only considers cold FEs [83]. Despite these conceptual differences, there is consensus that both aspects are involved in the integral development in individuals [84].

The most studied hot executive function would be gratification delay, which is defined by the willingness to access immediate rewards in favor of delayed gratifications of higher value [85]. Regarding the underlying neural mechanisms, as described in literature, limbic areas of the brain are associated as being sensitive to immediate rewards, while the lateral prefrontal cortex is considered as influencing the ability to delay gratification [86,87]. This neurocognitive assessment process involves two paradigms in a young child’s inclination to exhibit a gratification delay: delayed gratification choice and maintenance [46].

The second most studied hot executive function is decision making, despite being often classified as a cold or cognitive EF exclusively, is also involves emotional or hot components actively influencing this function [4]. Possible positive and negative outcomes or consequences, associated with a specific choice of activity, are considered in decision making [88]. This process is related to the somatic marker hypothesis, since actions are analyzed and organized by virtue of future outcomes, with positive or negative valence [24]. As described by Kable and Glimcher (2009), the performance of this task involves dorsolateral prefrontal areas for decision making, the amygdala to express emotional unpleasantness, basal ganglia to modulate behavior, the anterior cingulate cortex in order to relate amongst one another, and the nucleus accumbens in reward circuits, among others [89], where the IOWA Gambling Task (IGT) has been mostly used [69].

The third most studied hot executive function is self-regulation, in which evidence shows that the ventromedial and orbitofrontal cortex are associated as playing an active role in the processes of inhibition, emotion and reward elicitation [88]. Self-regulation is a complex and multifactorial concept [90] that acts at different executive functioning levels that represents the ability to voluntarily plan and modulate one’s behavior for an adaptive purpose [91]. Some authors consider the concept of behavioral self-regulation as synonymous with hot EFs [92,93], while others extend the concept, considering it as both hot and cold at the same time [94]. The Preschool Self-Regulation Assessment (PSRA) is used to evaluate hot and cold aspects where the focus is on the EF of delayed gratification [56,57,58,59].

Reward is a common factor in hot EF assessments although there is disagreement as to whether a hot EF task must include rewards, or some type of appetitive stimulus [95], or whether it only needs to elicit an emotional stimulus or increased motivation for it to be performed [16]. In relation to this, some authors propose that the EF performance depends on the influence that the motivational factor has on being either intrinsic or extrinsic [96].

According to reviewed evidence, hot EFs have implications on ADHD symptoms and behavioral problems [21]. They can be affected in adverse contexts, although improving when a training system is incorporated [47]. Hot EFs were not associated with the development of physical or relational aggression, nor significantly with Theory of Mind (ToM), socioeconomic adversity, cognitive self-regulation, or obesity [42,43,48,56]. There are no reported studies relating hot EFs and autism spectrum disorder (ASD) in preschoolers, though both concepts should be further studied. Such is due to the implication that this neurocognitive function has in personal and emotional interpretation, regulation of social behavior and understanding of social and contextual cues [97], all aspects that may be altered in ASD as currently conceptualized [98].

Favoring hot EFs, especially in the willingness to delay gratification, decision making and behavioral self-regulation, during early childhood predicts greater success academically and socially [99,100] and is to health and economic well-being in adulthood [101,102,103,104]. Hot EFs begin to develop early in life, enabling the development of inhibitory control, attention and working memory, forming the basis of voluntary control of behavior and thought, and the ability to interpret one’s own emotion as well as another’s, therefore enhancing the capacity for self-regulation and learning [5].

Consequently, longitudinal studies that consider problems such as addictions, behavioral disorders, or other pathologies where hot EF based brain structures are involved, are suggested so that predictive hot EF characteristics in preschoolers can be identified. It is also recommended that more studies and instrumental validations be carried out in both Latin American and Spanish-speaking countries, since there is little evidence of research in this population. The future of EF assessment faces the challenge of explaining the integration of cognitive and emotional components. Therefore, the evaluation of EFs should consider the type of stimulus or reward involved and must take into account the temperature of the cognitive activity.

Two limitations can be observed in the study, one being that these instruments are based on visual stimuli, thus requiring a preserved visual capacity, while not considering other sensory-perceptual pathways such as auditory, somesthetic, chemical, olfactory or gustatory. On the other hand, only four electronic databases were selected for the searches: EBSCO, WoS, SciELO and PubMed. Thus, it is possible that there are other articles available on the subject. In addition, the lack of conceptual unification of EFs limits the number of articles identified on the evaluation of hot EFs at a preschool age.

## 5. Conclusions

The most commonly used hot EF instruments in preschoolers are the Delay of Gratification Task (DGT), Gift Delay Task, Children’s Gambling Task and Preschool Self-Regulation Assessment (PSRA). Most of them assess delayed gratification, affective decision making and self-regulation with emotional components. The accuracy and conceptual clarity between the assessment of cognitive and emotional components in EFs is still debatable. The consideration of the affective temperature and the decision to choose the reward stimulus may have an impact on EF performance, situations that are suggested as involving limbic system structures, the lateral prefrontal cortex, ventromedial cortex, and the nucleus accumbens, among others. Likewise, the study of EFs integrated in their hot and cold aspects, can report a panorama of greater comprehensiveness in the cognitive-affective functioning of preschool children. However, the instruments are not designed with inclusivity criteria that considers other sensory-perceptual channels that may influence EFs. It is pertinent to continue studies in order to incorporate instruments from diverse information inputs into the executive system. The promising development of hot EFs in preschoolers can contribute to a diagnosis and design of rehabilitation strategies, allowing them to better perform in their educational and socio-cognitive life.

## Figures and Tables

**Figure 1 ijerph-19-00095-f001:**
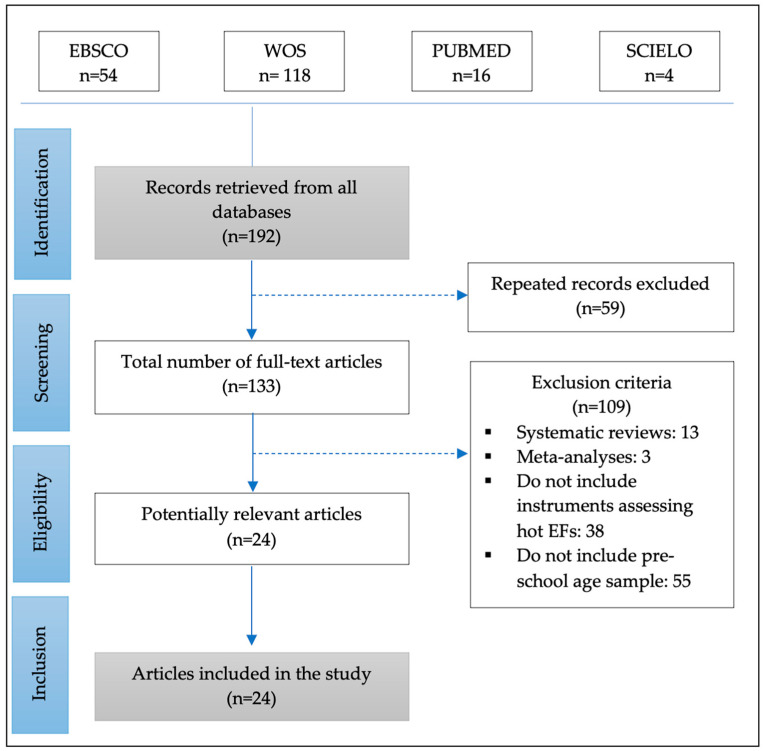
PRISMA Flowchart.

**Table 1 ijerph-19-00095-t001:** Hot EF instruments.

Instrument	Evaluated Function	Amount	Author	Psicometric Aspects
Less is More Task (LMT)	Inverse reward contingency	1	(Chi et al., 2018) [35]	Internal consistency = 0.9. Intraclass correlation Coefficient (ICC) = 0.97
Sticker Search (SS)	Decision making	1	(Chi et al., 2018) [35]	Internal consistency = 0.91. Intra class correlation Coefficient (ICC) = 0.97
Emotional Flexible Item Selection Task (EM-FIST)	Affective flexibility	1	(Martins et al., 2018) [36]	Not reported in this article
The Emotion Regulation Checklist (ERC)	Emotional regulation	1	(Martins et al., 2018) [36]	Significance level of α = 0.63
Iowa Gambling Task (IGT)	Decision making	1	(Garon y Longard, 2015) [37]	Not reported in this article
Children’s Gambling Task(ChGT)	Decision making	6	(Hongwanishkul et al., 2005 [18]; Kerr y Zelazo, 2004 [38]; O’Toole et al., 2017 [39], 2018 [40]; Poland et al., 2016 [41]; Putko, 2013 [42])	Not reported in this article
Delay of Gratification task (DGT)	Gratification delay	9	(Beck et al., 2020 [43]; Hodel et al., 2016 [44]; Hongwanishkul et al., 2005 [18]; Imuta et al., 2014 [45]; Mulder et al., 2014 [46]; Pellizzoni et al., 2019 [47]; Slot et al., 2017 [48]; Talwar et al., 2011 [49])	The test-retest reliability is 0.99. (Pellizzoni et al., 2019) [47]. Kappa (*n* = 53): 0.89 for tactile behavior and 0.74 for tearing the wrapping paper (Mulder et al., 2014) [46].
Gift Delay task (GDT)	Gratification Delay	6	(Montroy et al., 2019 [11]; O’Toole et al., 2018 [40]; Pellizzoni et al., 2019 [47]; Poland et al., 2016 [41]; Slot et al., 2017 [48]; Talwar et al., 2011 [49]).	The test-retest reliability is 0.97 for latency and 0.88 for violations (Pellizzoni et al., 2019) [47]. Latency to first glance (in seconds) was used in all analyses (ICC = 0.86–0.96), as it was available in all four studies (*n* = 1750) (Merz et al., 2014 [50], 2016 [51]; Sulik et al., 2010 [52]). Latency scores are highly correlated with other rating scores often derived from the task (r> 0.70) (Montroy et al., 2019) [10]. Reliability is tested in 10–20% of cases. K = 0.95 (Pauli-Pott et al., 2017) [53]. Factor loadings for each of the indicators of the latent constructs of hot executive functions were ≥ 0.77 and ≥0.41, respectively (all pags <0.001) (Slot et al., 2017) [48].
Dimensional Change card Sort (DCCS)	Flexiblity-Hot Version and Gratification delay	2	(Beck et al., 2011 [54]; Talwar et al., 2011 [49])	Overall same-day test-retest reliability (ICC = 75) on three of the tasks: Conflict-Cool, Conflict-Hot, and Delay-Hot. Delay-Cool, test-retest reliability did not meet psychometric standards (ICC 1⁄4 0.49) (Beck et al., 2011) [54].
Snack Delay Task (SDT)	Gratification delay	2	(Alesi et al., 2018 [55]; Slot et al., 2017 [48])	The factor loadings for each of the indicators of the latent constructs for hot executive functions were ≥ 0.77 and ≥0.41, respectively (all pags <0.001) (Alesi et al., 2018) [55].
Gift Wrap Task (GWT)	Gratification delay	4	(Alesi et al., 2018 [55]; Finch y Obradović, 2017 [56]; O’Toole et al., 2017 [39]; Pauli-Pott et al., 2017 [53])	Not reported in this article
Preschool Self-Regulation Assessment (PSRA)	Self-regulation	4	(Bassett et al., 2012 [57]; Denham et al., 2012 [58]; Finch y Obradović, 2017 [56]; Walczak y Chrzan-Dętkoś, 2018 [59])	The correlation coefficient between the two raters (two-way random model) in nine children was equal to 0.99 (Walczak y Chrzan-Dętkoś, 2018) [59]. Confirmatory factor analyses showed two components at each time point—hot and cold executive control—and cross-time correlations showed significant stability of individual differences (Bassett et al., 2012) [57]. (SR, α = 0.96) (Finch y Obradović, 2017) [56]
Teacher-reported (EFIn)	Child behavioral rating	1	(Montroy et al., 2019) [11]	Both scales were highly correlated, r = 0.76, and belong to the CBQ effort control factor (Rothbart et al., 2001) [60], subsequently averaged to create a single score (Sulik et al., 2010) [52].
Maudsley’s Index of Childhood Delay Aversion (MIDA) adapted version	Aversion to delay	1	(Hodel et al., 2016) [45]	High test-retest reliability among participants (Kuntsi et al., 2001) [61].
Cookie-Delay Task (CDT)	Gratification delay	1	(Pauli-Pott et al., 2017) [53]	Factorial and construct validity (Dalen et al., 2004 [62]; Pauli-Pott et al., 2014 [63]). Reliability is tested in 10–20% of cases. ICC = 0.99
Stranger-with-Toys (SwT)	Gratification delay	1	(Pauli-Pott et al., 2017) [53]	Reliability is proven in 10–20% of cases. ICC = 0.90

**Table 2 ijerph-19-00095-t002:** Hot EF Test Description.

Test	Author	Description
LMT	(Chi et al., 2018) [35]	(Carlson et al., 2005) [64]. The test has two levels: the first with 12 trials (choose between large and small candy tray). The second with 16 trials (two puppets, same rule and reverse reward). Duration: 18 min approximately.
SS	(Chi et al., 2018) [35]	(Choi y Song, 2013) [65]. The test uses 16 boxes with transparent lids, the child receives a reward when he or she selects the correct box.
EM-FIST.	(Martins et al., 2018) [36]	Version adapted from Mărcuş et al., (2015) [66]. The test presents 2 demonstration trials, 4 practice trials and 12 application trials; in them, children are shown cards with emotional and non-emotional characteristics.
ERC	(Martins et al., 2018) [36]	(Shields y Cicchetti, 1997) [67]. A 4-point Likert-type scale with 24 items. The ERC is composed of two different scales: a Negativity/Likability scale and the Emotion Regulation Scale.
IGT	(Garon y Longard, 2015) [37]	Children’s version of the Iowa Gambling Task. The administration of this task was inspired by Garon y Moore (2004) [68]. Sixty trials where children choose between two decks of cards and are told that the bear symbol would lead to winning a reward while the tiger symbol would lead to losing a reward, are presented.
ChGT	(Hongwanishkul et al., 2005 [18]; Kerr y Zelazo, 2004 [38]; O’Toole et al., 2017 [39], 2018 [40]; Poland et al., 2016 [41]; Putko, 2013 [42])	Simplified version of the Iowa Gambling Task (Bechara et al., 1994) [69] and adapted by Kerr y Zelazo (2004) [38]. Six demonstration trials and 50 test trials are presented, where children choose between two decks of cards and are told that the happy face corresponds to a reward while the sad face corresponds to the loss of a reward.
GDT o GWT	(Alesi et al., 2018 [55]; Finch y Obradović, 2017 [56]; Montroy et al., 2019 [11]; O’Toole et al., 2017 [39], 2018 [40]; Pauli-Pott et al., 2017 [53]; Pellizzoni et al., 2019 [47]; Poland et al., 2016 [41]; Slot et al., 2017 [48]; Talwar et al., 2011 [49])	(Carlson et al., 2005 [64]; Carlson y Moses, 2001 [70]; Kochanska et al., 1996 [71], 2000 [72]; Petersen et al., 2016 [73]). The task consists of telling children that they will receive a present, but that they cannot look at it while the experimenter noisily wraps the present, duration: one minute.
DCCS	(Beck et al., 2011 [54]; Talwar et al., 2011 [49])	Adapted from Zelazo (2006) [74]. Card sorting task, which in the hot version shows candies. First, it is sorted by shapes, where after 6 consecutive correct attempts, it is sorted by colors, then 12 trials are performed with instructions changing if a star appears on the card.
SDT	(Alesi et al., 2018 [55]; Slot et al., 2017 [48])	(Kochanska et al., 1996 [71], 1997 [75], 2000 [72]). The activity consists of showing children an attractive object, and then being asked to try not to touch it until the research assistant has completed another task.
PSRA	(Bassett et al., 2012 [57]; Denham et al., 2012 [58]; Finch y Obradović, 2017 [57]; Walczak y Chrzan-Dętkoś, 2018 [59])	(Smith-Donald et al., 2007) [76]. The PSRA is a battery composed of 10 tests that evaluate self-regulation. Within these tests, 4 correspond to evaluation of warm EFs, specifically delay of gratification (Toy Wrap, Toy Wait, Snack Delay and Tongue Task). It is performed through observing a child’s behavior, with interpretations that suggest the activation of areas in the nervous system.
EFIn	(Montroy et al., 2019) [11]	(Merz et al., 2014 [50], 2016 [51]; Sulik et al., 2010 [52]). This questionnaire is answered by teachers and is composed of two scales: 13-item attention concentration scale and 14-item inhibitory control scale from the Child Behavior Questionnaire (CBQ) (Rothbart et al., 2001) [57].
MIDA	(Hodel et al., 2016) [44]	Version adapted from Kuntsi et al., (2001) [61]. The task consists of a computer game set in a spaceship environment, where one must shoot asteroids to save a fictitious planet and receive rewards according to the time one waits to shoot.
CDT	(Pauli-Pott et al., 2017) [53]	(Carlson et al., 2005 [64]; Petersen et al., 2016 [73]). In this task, the child is instructed to wait for a bell to ring before he or she can retrieve a candy that is being covered by a transparent cup. Six trials are performed, plus one practice trial, with delay intervals between 10 and 40 s.
SwT	(Pauli-Pott et al., 2017) [53]	(Asendorpf, 1990 [77]; Pauli-Pott et al., 2014 [63]). In this task, the child sits at a table with an unappealing toy. A stranger enters the room, with interesting toys, and plays with them while not assisting the child. After 3 min, he invites the child to play with him along with the toys, for 2 min.
Among the Delay of Gratification Task (DGT) instruments, it is possible to distinguish three different tasks.		
DGT	(Beck et al., 2020 [43]; Hodel et al., 2016 [44]; Hongwanishkul et al., 2005 [17])	Prencipe y Zelazo Version (2005) [78] adapted from Thompson et al., (1997) [79]. The test presents 9 test types, created by crossing 3 reward types and 3 choice types (one now and two later, one now and four later, one now and six later), with 2 demonstration tests at the beginning. In the article by Hodel et al., (2016) [44] only 2 reward types are presented.
DGT	(Beck et al., 2020 [43]; Imuta et al., 2014 [45]; Talwar et al., 2011 [49])	(Mischel et al., 1989 [80]; Mischel y Ebbesen, 1970 [81]). Two rewards are presented, a small one (2 pieces) and a large one (10 pieces), the child being evaluated must wait to obtain the larger reward.
DGT	(Mulder et al., 2014 [46]; Pellizzoni et al., 2019 [47]; Slot et al., 2017 [45])	Adapted from Kochanska et al., (1996 [71], 2000 [72]). The activity consists of instructing the child to try not to touch the gift for a delay of 1 min.

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
