# Peer review of "Hot Executive Function Assessment Instruments in Preschool Children: A Systematic Review"

_ijerph, 2021, doi:10.3390/ijerph19010095_

Round 1
Reviewer 1 Report
Review Hot Executive Function Assessment Instruments in Preschool
Children: A Systematic Review
The definition of Executive Function offered by the authors, and the relationship of this function with the involvement of cortical structures and the limbic system, especially the dorsolateral prefrontal cortex and ventromedial frontal cortex, has contributed to interpreting that this research was situated in the field of the neurosciences.
In a second reading we have already been able to locate the research in the field of psychology or applied psychology. This needed to be clarified from the beginning in order not to confuse the non-expert. The batteries and tests referred to in the article do not include brain imaging techniques to visualize the activity that occurs within the brain during a certain activity or behaviour, as for example occurs in the research “Behavioural and neural correlates of delay of gratification 40 years later” de Casey, Somerville, et al. (2011) in which they were using functional magnetic resonance imaging (fMRI). Without neurological instruments, the involvement of cortical structures and the limbic system, especially the dorsolateral prefrontal cortex and ventromedial frontal cortex, we do not see how it can be evaluated.
Then, this research maintains its great specificity. The topic focuses on inquiring which instruments are most used to assess executive function in preschool. The research is briefly presented. Which hurts non-PE experts. The conceptual foundation on EF is limited to 37 narrow lines, the assessment to 8 lines. The methodology of selection and review of articles 20 lines without explaining the PRISMA instrument used. The presentation of results basically consists of two tables. In table 1 the instruments are named but not explained. The evaluated function is named but also not explained. In sum, the presentation of results focuses on table 1 and table 2 with a very brief the additional information what forces the no EF expert to interpret the tables.
There is little information on drums, test or instruments, not even the most used according to the results as Delay of Gratification Task, the Children’s Gambling Task, and the Gift Delay Task. After browsing Google we can check the functioning of the psychological instruments by ourselves. For instance, The PSRA Assessor Report, a developed by Raver and colleagues, is designed to capture assessors' ratings of children's emotion regulation and attention/impulsivity during the course of the PSRA administration but it does not imply a neurological evaluation.
Arguably there is necessary information that is not offered. In the discrete discussion and conclusions we do not understand why the authors return with neurological references if the research has focused on reviewing articles to know the most used psychological instruments in the delay of gratification. What proceeded was to delve further into the justification of the selection and review of articles. And better inform about the types and functions of the instruments.

Author Response
Dear reviewer,
Thank you for your comments, they have helped us to clarify our article and better it's content. We have responded to each of the paragraphs you sent, indicating the changes in the text.
Paragraphs 1 and 2
The definition of Executive Function offered by the authors, and the relationship of this function with the involvement of cortical structures and the limbic system, especially the dorsolateral prefrontal cortex and the ventromedial frontal cortex, has contributed to interpret that this research was located in the field of neurosciences.
In a second reading we have already been able to situate the research in the field of psychology or applied psychology. This should have been made clear from the beginning so as not to confuse non-experts. The batteries and tests referred to in the article do not include brain imaging techniques to visualize the activity that occurs within the brain during a given activity or behavior, as for example occurs in the research "Behavioural and neural correlates of delay of gratification 40 years later" by Casey, Somerville, et al. (2011) in which they used functional magnetic resonance imaging (fMRI). Without neurological instruments, the involvement of cortical structures and the limbic system, especially the dorsolateral prefrontal cortex and ventromedial frontal cortex, we do not see how it can be assessed.
R1. Regarding the context. In the first paragraph of the manuscript we contextualized the study based on executive functions within the framework of Neuropsychology as a discipline of Neuroscience and its object of study, whose interpretation is based on findings at the neurological level. A paragraph was added to the introduction:
"This study is framed in the context of Neuropsychology, a clinical discipline belonging to a part of neuroscience that focuses on a neural interpretation of cognitive-behavioral and affective evidence in people [1,2]. In this line, memory, attentional and executive functions (EF), which is the particular interest of this study, are evaluated according to the person's response in relation to a possible neurological correlate. "
Regarding the quote referring to Casey, Somerville, et al. (2011), Table 2 of the manuscript provides a more detailed explanation of the PSRA battery and refers to the literature that supports it. In the other reward delay tests, interpretations are also made that allude to or suggest the involvement of cortical or subcortical areas, typical of the discipline of Neuropsychology.
Paragraph 3 subdivided:
Then, this research maintains its great specificity. The topic focuses on investigating which instruments are the most commonly used to assess executive function in preschoolers. The research is briefly presented. What is detrimental to non-experts in EF. The conceptual foundation on EF is limited to 37 narrow lines, the assessment to 8 lines.
R2. Two paragraphs have been added at the end of each table, in order to clarify the details of its content. The added texts are the following:
(i) "The list of instruments that were considered, together with the function assessed and the psychometric aspects that account for reliability and validity aspects, are shown in Table 1. It can be seen that some studies refer exclusively to hot FE aspects and others to the combination of cold and hot FE. "
ii)" As can be seen in Table 2, the DGT test and the PSRA battery stand out as the most widely used instruments in the measurement of delay of gratification since they involve consideration or waiting for a greater reward. Others, such as the CDT and SDT, wait for the sound of a bell to receive a reward. While, with computational applications, games that include rewards, as opposed to waiting for gratification, have also been used. An example of this is MIDA, most commonly used with children over 5 years old. Gestures made when approaching a more entertaining toy are also measured using tests such as the SwT. Questionnaires are also used that are completed by outside observers, such as teachers or parents, in which they check the gratification control of access to EFIn. Others address attentional inhibition to a desirable object while performing tasks, which can be seen in tests such as the DCCS and GDT. The ERC and EM-FIST tests assess affective flexibility in performing a task with the influence of a significant character. There are also those in which rewards are given when the less advantageous one is chosen, such as the LMT and SS, a situation referred to as reverse reward in the case of the former. The IGT and ChGT use cards in which people evaluate the ones with something advantageous, such as smiley faces. "
The methodology of selection and review of articles 20 lines without explaining the PRISMA instrument used.
R3. The PRISMA methodology is briefly explained for any systematic review or meta-analysis with precise and consensual standards for publication. The following text is added in the Methodology section:
:The review of articles is based on the Preferred Reporting Items for Systematic Reviews and Meta-Analyses (PRISMA) considering selected articles that address the effect of interventions on the topic of hot EF [33,34]. "
Paragraph 4.
There is little information on drums, tests, or instruments, not even the most commonly used ones according to the results such as the Delayed Gratification Task, the Infant Play Task, and the Delayed Gift Task. After browsing Google we can check the performance of the psychological instruments for ourselves. For example, the PSRA Assessor Report, developed by Raver and colleagues, is designed to capture assessors' ratings of children's emotion regulation and attention/impulsivity during the course of PSRA administration but does not involve neurological assessment.
R4. Regarding the brain structures, the wording that can be seen below in the manuscript was added when mentioning structures:
"Regarding the underlying neural mechanisms, as described in the literature, limbic areas of the brain are associated as sensitive to immediate rewards, whereas the lateral prefrontal cortex is considered to influence the ability to delay gratification [86,87]. This neurocognitive assessment process implicates two paradigms in a young child's inclination to show delayed gratification: delayed gratification choice and maintained "
Paragraph 5.
Arguably, there is necessary information that is not offered. In the discrete discussion and conclusions, we do not understand why the authors return with neurological references if the research has focused on reviewing articles to learn about the psychological instruments most commonly used in delayed gratification. What was appropriate was to go deeper into the rationale for the selection and review of articles. And to better inform about the types and functions of the instruments.
R5. Neurological references in neuropsychology usually occur since neuropsychology is a discipline that examines the responses or behaviors of individuals and offers an interpretation according to the neural processes in cortical and subcortical structures. Key aspects are added in the discussion and conclusions in the manuscript, considering this explanation.
Reviewer 2 Report
Thank you for the opportunity to review the manuscript “HOT EXECUTIVE FUNCTION ASSESSMENT INSTRUMENTS IN PRESCHOOL CHILDREN: A SYSTEMATIC REVIEW”.
The manuscript is interesting because it provides insight into the situation of the empirical evidence concerning instruments for assessing executive functions in pre-schoolers. It is an enjoyable text to read, well structured and very interesting. However, its scientific content is not too deep since it does not delve too deeply into executive function in early childhood, but this is not a problem from my point of view for it to be accepted in the journal. So, the introduction could be expanded to delve into aspects of other similar studies as well as the subject under study in early childhood.
The abstract can be improved, better reflecting the different sections of the work.
The discussion and conclusions sections are too poor and should be expanded. It could be developed by commenting more on the main results, offering certain opinions of the authors themselves on the situation in Spanish- speaking countries as well as in relation to previous studies. Finally in this section, it would be interesting to offer suggestions and lines of research for future studies that other authors or themselves could develop, especially from an empirical perspective.
Given that the approach and the contents of the manuscript are of interest as it allows future investigation of the reality of early childhood in Spanish-speaking countries, my opinion is that the article can be accepted for publication in the journal, after slight changes and modifications.
Finally, I want to thank again for having the opportunity to read the work and I encourage the authors to continue working in this line.
Author Response
Dear Mr. Reviewer.
We thank you for your comments and respond below to your feedback for each paragraph:
Paragraph 1. The manuscript is interesting because it provides information on the status of empirical evidence on instruments for assessing executive functions in preschoolers. It is a pleasant text to read, well structured and very interesting. However, its scientific content is not too deep as it does not go too much into executive function in early childhood, but this is not a problem from my point of view for it to be accepted in the journal. Thus, the introduction could be expanded to delve into aspects of other similar studies as well as the topic under study in early childhood.
R1. A text contextualizing executive functioning is added to the introduction:
This study is framed in the context of Neuropsychology, a clinical discipline pertaining to a part of neuroscience that focuses on a neural interpretation of the behavioral and affective cognitive evidence in people [1,2]. Along these lines, the mnesic, attentional and executive functions (EFs), which is the particular interest in this study, are evaluated according to a person‘s response in relation to a possible neurological correlate.
The age range is incorporated into the discussion by alluding to the context of neuropsychological assessment. The párrafo is transcript here:
Favoring  hot  EFs, especially in the willingness to delay gratification, decision making, and behavioral self-regulation, during early childhood predicts greater success academically and socially [99,100] as well as being related to health and economic well-being in adulthood [101-104].  Hot executive functions begin to develop early on, enabling the development of inhibitory control, attention, and working memory, forming the basis of voluntary control of behavior and thought, and the ability to interpret one's own emotion, as well as another, therefore enhancing the capacity for self-regulation and learning [5].
Paragraph 2. The abstract could be improved by better reflecting the different sections of the paper.
R2. The abstract was rewritten:
Paragraph 3. The discussion and conclusions sections are too poor and should be expanded. They could be developed by commenting more on the main results, offering certain opinions of the authors themselves on the situation in Spanish-speaking countries as well as in relation to previous studies.
R3. They were fundamentally expanded, although the deepening of some topics are already considered in the references. Therefore, we emphasize the fundamentals.
Párrafo 4. Finally in this section, it would be interesting to offer suggestions and lines of research for future studies that other authors or they themselves could develop, especially from an empirical perspective.
R4. They were included in the conclusions, which we transcribe the aggregate:
However, the instruments are not designed with inclusivity criteria that considers other sensory-perceptual channels that may influence EFs. It is pertinent to continue studies in order to incorporate instruments from diverse information inputs into the executive system. The promising development of hot executive functions in preschoolers can contribute to a diagnosis and design of rehabilitation strategies, allowing them to better perform in their educational and socio-cognitive life.